# Effectiveness of Non-Contact Dietary Coaching in Adults with Diabetes or Prediabetes Using a Continuous Glucose Monitoring Device: A Randomized Controlled Trial

**DOI:** 10.3390/healthcare11020252

**Published:** 2023-01-13

**Authors:** Yeh-Chan Ahn, Yang Seok Kim, Bukyung Kim, Jung Mi Ryu, Myoung Soo Kim, Minkyeong Kang, Jiwon Park

**Affiliations:** 1Department of Biomedical Engineering, Pukyong National University, Busan 48513, Republic of Korea; 2Department of Physiology, College of Korean Medicine, Kyung Hee University, Seoul 02447, Republic of Korea; 3Division of Endocrinology and Metabolism, Department of Internal Medicine, College of Medicine, Kosin University, Busan 49267, Republic of Korea; 4Department of Nursing, Busan Institute of Science and Technology, Busan 46639, Republic of Korea; 5Department of Nursing, Pukyong National University, Busan 48513, Republic of Korea

**Keywords:** diabetes mellitus, prediabetic state, healthy lifestyle, diet, continuous glucose monitoring

## Abstract

We aimed to evaluate the effectiveness of dietary coaching and continuous glucose monitoring (CGM) in patients with diabetes or prediabetes to improve their behavioral skills and health outcomes. A randomized controlled study with pre- and post-testing was conducted. Data were collected between November 2020 and April 2021. Forty-five patients with diabetes or prediabetes who used a CGM device were enrolled and analyzed. Dietary education, individual coaching and group coaching were provided to participants in the experimental group for 4 weeks. After the intervention, the thigh circumference in men significantly differed between the two groups (*z* = −2.02, *p* = 0.044). For women, participants in the experimental group showed greater improvement in eating self-efficacy compared with those in the control group (*z* = −2.66, *p* = 0.008). Insomnia was negatively related to the change in eating self-efficacy (*r* = −0.35, *p* = 0.018) and increase in thigh circumference (*r* = −0.35, *p* = 0.017). Even if used within a short intervention period, non-contact dietary coaching programs can help enhance behavioral skills, such as eating self-efficacy and health outcomes, such as thigh circumference. Moreover, the changed variables can indirectly improve other health outcomes in patients with diabetes or prediabetes.

## 1. Introduction

The global prevalence of type 2 diabetes mellitus (T2DM) is gradually increasing. In a recent study [1], Korean adults aged ≥30 years showed a prevalence rate of 13.8%, which is relatively high compared with the global adult prevalence rate of 9.3% as of 2019 [2]. Approximately 26.9% of adults aged ≥30 years have impaired glucose metabolism and the proportion of this patient group is increasing rapidly worldwide [1]. Prediabetes indicates that an individual may have more than one sign of impaired glucose metabolism, including abnormal fasting glucose levels, impaired glucose tolerance and glycated hemoglobin (HbA1c) levels that are above normal but below diabetes-associated levels [3]. For patients with prediabetes, early detection and prompt intervention are important, as the condition may be reversible through lifestyle changes.

To prevent T2DM and improve glycemic control, multiple lifestyle modifications are recommended, including reducing weight, increasing physical activity, adopting low-saturated fat and high-fiber diets and performing self-monitoring [4,5]. Not only can these interventions prevent the transition to T2DM in patients with prediabetes, but they can also delay the onset of complications in those with diabetes [6]. Among these, dietary modification is the key intervention for diabetes [7]; the use of digital devices to facilitate education and behavior change and participation in online coaching are especially effective [8]. Although there is no clear evidence to support that diet or physical activity alone reduces the risk of developing complications in T2DM [9], the fact that dietary education alone, when provided for at least 3 months, is effective in managing the HbA1c levels in people with T2DM [10] indirectly demonstrates the importance of dietary coaching. A healthy diet, with or without physical activity, reduces the risk of T2DM [11]. Furthermore, encouraging patients to perform continuous glucose monitoring (CGM) and to adhere to a low-carbohydrate diet can boost their satisfaction level, reduce weight [12] and decrease HbA1c levels [13].

CGM involves the use of a minimally invasive device equipped with a tiny sensor electrode that is inserted underneath the skin to monitor variations in blood glucose levels [14]. CGM devices provide glucose readings every 5 min, record fluctuations in glucose levels and sound an alert to warn users of low or high glucose levels. Two types of devices are currently available: traditional CGM and flash CGM. The former passively transmits glucose information continuously, without user involvement, whereas the latter provides information when the user scans the sensor [15,16]. As such, the use of flash CGM elicits greater engagement from participants. To date, most studies investigating the effectiveness of CGM for glucose management have been conducted in individuals with type 1 diabetes mellitus [17]. Compared with traditional blood sugar self-monitoring, CGM is effective for improving HbA1c levels [18], glucose variability [19] and self-care practices [20] in patients with T2DM. Although the benefits of CGM for T2DM are not well established [21], CGM has recently been included in the professional guidelines and standards of care for T2DM [22]. Consistent with this, CGM use increases the efficacy and reduces the overall costs of T2DM management [22].

Several studies have performed lifestyle modification interventions in populations with T2DM [23,24], adapting the information–motivation–behavior (IMB) model [25]. The IMB model includes six factors: information, personal and social motivation, behavioral skills, behaviors and physiological outcomes. In accordance with the IMB model, healthy dietary behavior necessitates gaining access to good information, is prompted by strong individual and social motivation and requires behavioral skills necessary for adherence [25]. In an IMB context, CGM can be viewed as an educational [26] and motivational tool to facilitate patients’ monitoring of their blood glucose levels [27]. Therefore, as a relevant component of a dietary coaching program, CGM may positively affect both behavioral skills and health outcomes. However, as most studies to date have assumed that all lifestyle factors simultaneously change during an intervention, the nature of an interim cascade of effects is unknown. For example, a change in behavioral skill is a procedural change that may have resulted from the acquisition of new information or improvements in personal or social motivation. In this phase of the IMB model, self-efficacy [28] is the variable of interest. In turn, new healthy behaviors may alter psychological factors, such as depression, concentration level and insomnia [29]. Consequently, improvements in these domains may yield favorable health outcomes by affecting anthropometric parameters, such as body mass index (BMI), waist circumference [30], weight [31] and muscle strength, as well as biochemical indices such as glucose levels and insulin sensitivity [30,32]. However, previous studies have yielded inconsistent results and the factors that can be considered as the primary variables of change in the IMB model remain unclear.

To date, there is no clear clinical evidence that confirms the efficacy of dietary coaching using CGM devices in patients with T2DM or prediabetes and only a few studies have examined the relevant variables based on a theoretical framework. The data obtained from CGM biosensors can be used to develop a more detailed diet coaching program and an artificial intelligence diet-suggestion program for reducing the blood glucose level. If a relationship between changes in behavioral skills and health outcome variables after intervention is identified, the indirect effects of the variables can be utilized to develop more effective interventions. Thus, we aimed to evaluate the effectiveness of CGM combined with dietary coaching in patients with diabetes or prediabetes in order to improve behavioral skills and health outcomes. Based on the conceptual framework, two hypotheses were formulated. First, we hypothesized that male and female participants in the intervention group who adopted the diet coaching program would show enhanced behavioral skills (eating self-efficacy) and better health outcomes (improved depression and insomnia status, anthropometric parameters and biochemical indices) in the post-test compared with those in the control group. Second, we speculated that post-intervention changes in behavioral skill and health outcomes would be correlated with each other.

## 2. Materials and Methods

### 2.1. Study Design

This study was a 4-week, randomized controlled trial.

### 2.2. Participants and Recruitment

The participants were divided into control and experimental groups. Patients with diabetes or prediabetes from one outpatient department of Endocrinologic Internal Medicine and one obesity clinic in a private tertiary hospital, respectively, were recruited. The Republic of Korea has established a healthcare system that is funded under the National Health Insurance Service, a public medical insurance program run by the Ministry of Health and Welfare. Therefore, the level of health insurance coverage of all participants was similar. Patients aged 18–70 years, diagnosed with diabetes (HbA1c of ≥6.5%) or prediabetes (HbA1c of 5.7–6.4%), with a BMI of ≥23.5 kg/m^2^ who used medications to control blood glucose levels, who used a CGM device and who provided consent were included in the study. By contrast, patients who underwent bariatric surgery within the past 6 months, received chemotherapy within the past 2 months and showed signs of cognitive impairment were excluded. A total of 520 patients with diabetes and prediabetes met the eligibility criteria and were assigned to the diabetes and prediabetes groups in a 2:1 matching ratio.

The required sample size was estimated using the G*Power 3.1.9.4 program (Heinrich Heine University, Dusseldorf, Germany). A large effect size (effect size = 0.76), a typical significance level (alpha = 0.5) and a power of 0.8 based on a previous study [33] were the input conditions. Consequently, the required sample size was 25, with an estimated dropout rate of 20% in each group. The allocation of patients into the two study groups was carried out based on the following two processes: assignment and then concealment of the assignment. To guide the initial patient assignment, computer-assisted randomization was performed to assign patients to either the experimental or control group, in a 1:1 ratio. Next, one research coordinator created a set of sealed envelopes with allocation information and provided them to clinicians as patients visited their customary outpatient department. The clinicians explained the research protocol to each patient. Another research coordinator enrolled the patients in this study and performed pre-test and post-test assessments. Both patients and clinicians were blinded to the treatment assignment.

Initially, 25 patients were enrolled in each group; three patients from the control group and two patients from the intervention group subsequently withdrew from the study (Figure 1). The study began in November 2020 and was discontinued in April 2021 because of study duration limitations.

### 2.3. Intervention

All participants in the study used the Abbott FreeStyle Libre CGM software (https://www.freestyleprovider.abbott/us-en/home.html (accessed on 20 November 2020)). The FreeStyle Libre CGM system consists of a sensor that is attached to the back of the upper arm and inserted below the skin [15]. Each sensor is calibrated and can be used for up to 2 weeks. A user device or smartphone with near-sensor communication capability can be shown at any time to scan the sensor. The device or smartphone monitor shows the current glucose level, the glucose level trend and a graph of the glucose levels obtained in the last 8 h [16]. The CGM results including the current glucose level, the glucose trend and the graph were not used as one of the biochemical indices in this study. This is because it was not possible to accurately analyze the effects of this program as the participants’ diet and medication intake time were unknown. In addition to CGM, the control group received conventional care involving self-administered management, including education on the necessity of physical activity and compliance with the medications prescribed by the treating physician. Consultations and pamphlet-based education were provided during this study and conducted for 5–10 min per clinic visit at the start of treatment and after 2 weeks. Behavioral skills and health outcomes were assessed after 4 weeks when the patients visited the outpatient department.

The experimental group received two types of interventions: conventional care and a non-contact, nurse-led dietary coaching program. Based on the IMB model, the dietary coaching program consisted of education, dietary feedback with individual coaching and group coaching. Education was conducted every Monday, dietary feedback and individual coaching were conducted daily and group coaching was conducted every Thursday. Three trained nurses executed the coaching program. Prior to interaction with the participants, the nurses underwent three training sessions to learn how to calculate the amount of food and calories consumed by each patient; the training continued until an inter-class coefficient of 0.7 was reached.

The educational content was created by one of the researchers and included narrated PowerPoint presentations with recorded explanations of key points. The content was discussed in four sessions, with each session lasting 15–20 min. During individual coaching, the participants were advised to plan a healthy diet, including an appropriate caloric intake; appropriate ratios of carbohydrates, proteins and fats; and an appropriate glycemic index target according to their height and weight. They were also encouraged to send, on a daily basis, photographs and written descriptions of their actual dietary intake, including food names and quantities, to the nurse in charge. Any snacks or beverages consumed were also included in the dietary intake. The nurses analyzed the data indicated in patients’ diet diaries using the Computer-Aided Nutrient Analysis Program 5.0 (The Korean Nutrition Society, Seoul, South Korea) and provided dietary feedback to patients via text messages.

The Goals, Reality, Options and Will (GROW) model was used to construct an individual coaching content. The nurses provided individual coaching based on the nutrient analysis results, including daily dietary goals (G), achievements (R) and willpower (W) [34]. The educational content used for individual coaching was sent as text messages. The participant took 5–10 min to document their daily diet and receive feedback. Group coaching was conducting via a non-face-to-face group interaction platform; the participants were distributed into three groups, with each group comprising seven to eight patients. Each session lasted 30 min. Figure 2 shows the conceptual framework of the data collection process, including the outcome measures.

### 2.4. Outcome Measures

#### 2.4.1. Behavioral Skills

Eating self-efficacy: To assess for weight efficacy, which is a critical factor for the successful maintenance of weight management behavior [35], we used the Weight Efficacy Lifestyle Short-Form questionnaire. This is an 8-item questionnaire and the items are scored from 0 (not confident) to 10 (very confident), with higher scores indicating greater self-efficacy in controlling eating behavior. Cronbach’s alpha value was 0.83 in a previous study [35] and 0.80 in this study.

#### 2.4.2. Health Outcomes

Depression: Depression was measured using the Korean version of the Patient Health Questionnaire-2, which measures the frequency of depressed mood and anhedonia [36]. An example item asked about depressed mood (“How often have you been bothered by depressed mood (or anhedonia) over the last 2 weeks?”) and was scored as 0 (not at all), 1 (several days), 2 (more than half the number of days), or 3 (nearly every day). A higher score indicates more severe depression. Cronbach’s alpha value was 0.66 in a previous study [36] and 0.77 in this study.

Insomnia: To assess for presence of insomnia, we used the Korean version of the 4-item Mini-Sleep Questionnaire-Insomnia [37]. This questionnaire is used to examine the difficulties in initiating and maintaining sleep, instances of waking too early and the use of sleep medications; the items are rated on a 7-point scale, with scores ranging from 1 (never) to 7 (always). A high score indicates severe insomnia. In another study, Cronbach’s alpha was 0.69 and the test–retest reliability was 0.84 [37]. In this study, Cronbach’s alpha was 0.82.

Anthropometric parameters: The anthropometric parameters in this study were BMI and waist and thigh circumferences. A scale was used to determine the patients’ weight and tape measures were used to assess the waist and thigh circumferences.

Biochemical indices: The biochemical indices were HbA1c level, apolipoprotein B (ApoB)/apolipoprotein A (ApoA) ratio and the Homeostatic Model Assessment for Insulin Resistance (HOMA-IR) score. HbA1c level was calculated using the Nathan formula. Serum ApoA levels were determined using the Human ApoA1 enzyme-linked immunosorbent assay (ELISA) kit (no. 3710-1HP-2; Mabtech, Cincinnati, OH, USA), while serum ApoB levels were determined using the human ApoB ELISA kit 3715 (no-1HP-2 Mabtech, Cincinnati, OH, USA). A polyethylene glycol-enhanced immunology turbidimetric assay was performed to measure the ApoB/ApoA ratio using a 7600-010 automatic analyzer (Hitachi, Ibaraki, Japan). A high ApoB/ApoA ratio, indicative of insulin resistance and abnormal lipid metabolism, is a known risk factor for cardiovascular disease. The HOMA-IR score is used for assessing insulin resistance and is calculated based on fasting insulin and fasting blood glucose levels with the HOMA Calculator v2.2.3 (https://www.dtu.ox.ac.uk/homacalculator/ (accessed on 19 November 2020)).

### 2.5. Ethical Considerations

This study was approved by the Institutional Review Board (IRB no. 1041386-202011-HR-64-01) of the Pukyong National University in Busan, Korea. During the enrollment stage, the purpose of the study, voluntary nature of participation, confidentiality of information and study procedures were explained to potential participants. Written informed consent was obtained from each participant prior to data collection.

### 2.6. Data Analysis

The data were analyzed using IBM SPSS software for Windows version 27.0 (IBM SPSS Corp. Armonk, NY, USA). Descriptive statistics, t-tests, chi-squared tests, Fisher’s exact tests and Mann-Whitney U tests were used to evaluate the patients’ demographic data and the characteristics of the research variables. To identify the relationship between changes of the outcome variables, Pearson’s correlations were performed.

## 3. Results

The demographic characteristics of the two groups were homogeneous (Table 1). After the intervention, the thigh circumference of male participants was significantly different between the two groups (*z* = −2.02, *p* = 0.044, effect size = 0.255) (Table 2). However, no significant differences were observed in the anthropometric variables, including BMI (*z* = −1.56, *p* = 0.118) or waist circumference (*z* = −1.01, *p* = 0.350), at post-testing. In terms of the biochemical indices, no significant changes were observed in HbA1c levels (*z* = −0.31, *p* = 0.754), ApoB/ApoA ratio (*z* = −0.30, *p* = 0.763), or HOMA-IR scores (*z* = −0.10, *p* = 0.920) between the experimental group and control group. Among female participants, eating self-efficacy significantly improved in the experimental group compared with that in the control group after the intervention (*z* = −2.66, *p* = 0.008, effect size = 0.262) (Table 3). However, no significant differences were observed in the health outcome variables between the experimental and control groups, including depression status (*z* = −0.44, *p* = 0.660), insomnia status (*z* = −1.34, *p* = 0.180), BMI (*z* = −0.14, *p* = 0.887), waist circumference (*z* = −1.28, *p* = 0.200), thigh circumference (*z* = −1.63, *p* = 0.104), HbA1c level (*z* = −1.02, *p* = 0.308), ApoB/ApoA ratio (*z* = −153, *p* = 0.126), or HOMA-IR score (*z* = −0.07, *p* = 0.944).

Insomnia was negatively associated with change in eating self-efficacy (*r* = −0.35, *p* = 0.018) and increased thigh circumference (*r* = −0.35, *p* = 0.017), while change in waist circumference was positively associated with increased thigh circumference (*r* = 0.30, *p* = 0.048) and change in ApoB/ApoA ratio (*r* = 0.35, *p* = 0.020) (Table 4).

## 4. Discussion

This study investigated the sole effect of dietary coaching when patients with diabetes or prediabetes were already performing CGM, which is noteworthy considering that most previous studies have focused only on evaluating the effectiveness of CGM compared with that of conventional care. After the intervention in the experimental group, only the thigh circumference of the male participants and eating self-efficacy of the female participants showed significant improvements compared with those in the control group. Therefore, this finding partially supports our first hypothesis. Moreover, changes in behavioral skills and some of the health outcome variables were correlated to each other, partially supporting our second hypothesis. Here, characteristics of the dietary coaching program that led to the positive changes in thigh circumference and eating self-efficacy will be discussed first, followed by an analysis of the relationship among significant changes in patients’ scores.

Among male participants, the experimental group showed a significant improvement in thigh circumference compared with that in the control group. Although no significant differences were observed during the 4-week study period, BMI decreased by 1.72 kg/m^2^, waist circumference decreased by 4.50 cm and thigh circumference increased by 2.37 cm in male participants from the experimental group. These results are similar to those reported in patients with diabetes after 12 weeks of dietary intervention, which reported a decrease of 3.1 kg in body weight and an increase of 3.1 cm in waist circumference [38]. Similarly, positive changes were observed in body weight and waist circumference among patients with T2DM and abdominal obesity after adopting a lifestyle improvement program [39,40]. Although it is difficult to make a direct comparison between the results of our study and those of other studies owing to the scarcity of studies investigating changes in thigh circumference, education about physical activity may be effective in improving muscle mass among male participants who are relatively active and in good physical condition. If this trend had been maintained for another 8 weeks, a significant change in scores would have been observed.

Among female participants, only eating self-efficacy showed a significant difference between the intervention and control groups. Eating self-efficacy is an important predictor of weight-loss behaviors [41] such as adhering to one’s dietary plan and limiting calorie intake. In a recent study, a structured lifestyle program improved the eating self-efficacy in patients with T2DM and overweight/obese who maintained their glycemic level [42]. For those with gestational diabetes mellitus, telehealth coaching increased their ability to improve their diet [43]. In our study, coaches used several strategies based on Bandura’s four sources of general self-efficacy [44]: mastery experience, vicarious experience, verbal persuasion and physiological and affective states. To obtain a mastery experience, coaching and individual feedback were provided to patients to inform them if their daily diets were successful or unsuccessful based on the photographic records of meals. Sharing personal experiences during group coaching provided patients with vicarious experience. Individual coaching involved verbal persuasion to affirm participants of their capabilities. Lastly, administering knowledge tests and offering rewards during group coaching sessions and individualized testing periods improved patients’ physiological and affective states. These factors may have led to the improvements in eating self-efficacy among women.

In the dietary coaching program development phase, we predicted that only the implementation of CGM could not only improve the glycemic control [45] and glucose variability, including HbA1c level [46], but also reduce the psychological problems, such as depression, emotional burden [45] and glycemic diabetes distress [47]. Furthermore, because dietary coaching in this study was provided using a non-contact method, a stronger coaching program is warranted. To overcome the limitations of the non-contact method, we developed a tighter dietary coaching program based on the GROW model [34], which is the most established coaching model. During the period of individual and group coaching, coaching enabled the participants to establish their diet modification goal (G), including total calories, weight and glucose levels; examine their current reality in terms of achieving their goal (R); determine their options (O) to share experiences (via social network site); and establishing the will (W) to commit to specific actions toward achieving their goal. However, this might have made participants in the experimental group feel more suppressed compared with those in the control group. No studies have focused solely on treating depression in the coaching program; however, a previous study reported the negative relationship between depression and dietary adherence [48]. Although no significant difference was observed, the difference scores slightly increased during post-tests compared with that during pre-tests, thus raising concerns about participants’ diet adherence. Therefore, flexibility in adapting the model should be considered not only during the development stage, but also during the implementation stage of the coaching program.

In this study, the absence of differences in several physical health outcomes between the two groups can be attributed to two factors. First, CGM itself is a particularly effective intervention. CGM has been utilized to provide biofeedback, monitor dietary adherence and assess glycemic variability [49]. According to several systematic reviews, CGM is more effective in reducing HbA1c level and hypoglycemic events compared with self-glucose monitoring in T2DM patients [33] and pregnant women [50], can provide more accurate glycemic data in T2DM patients on hemodialysis [51] and can improve the diabetes-related distress level, family conflicts and quality of life [52]. Therefore, the changes induced by dietary coaching may have been offset by the potentially more significant effects of CGM. Second, the non-contact approach to coaching could have reduced the effectiveness of our intervention. Although non-contact methods may be more convenient, patients may benefit more from direct contact with dietitians [53]. In addition, some participants may feel uncomfortable receiving dietary coaching through a social network service platform. Hence, revisions to the intervention program that can enhance direct contact with coaches should be considered.

Insomnia was negatively related to eating self-efficacy and reducing thigh circumference. As previously mentioned, eating self-efficacy is an important predictor of weight loss behavior. Participants with high eating self-efficacy are likely to maintain adaptive weight loss behavior to achieve and sustain weight loss. Based on a recent study, greater weight loss, fat loss and fat-free mass loss were associated with improved sleep health [54]. Increased thigh circumference may indicate increased muscle mass, which can be helpful for improving sleep quality. Patients with insomnia, for example, are more likely to experience reduced muscle mass [55] and a higher muscle mass can be related to earlier wake time and shorter sleep latency [56]. Simply put, improvement in behavioral skill (such as eating self-efficacy) and anthropometric health outcomes (such as muscle thickening) can improve sleep quality. In our study, waist circumference was related to the ApoB/ApoA ratio; a positive correlation was also found between waist circumference and insulin resistance, insulin levels and HbA1c levels [57]. Other studies reported that ApoB/ApoA ratio is correlated with the risk of myocardial infarction and carotid intima-media thickness in patients with T2DM [58]. Furthermore, the incidence of T2DM is 6.92 and the new-onset T2DM risk ratio is 2.42 in patients with high ApoB/ApoA ratio trajectories compared with those with low ApoB/ApoA ratio trajectories [59]. Given that the ApoB/ApoA ratio is associated with risk of diabetes and prediabetes [60], reduction of waist circumference may be a useful strategy in this patient group.

Although it provided meaningful findings, this study has several limitations. First, the 4-weeks intervention period may not be sufficiently long to change several outcome variables. When the coaching program was developed, we referred to a systematic review in which 23 (81.1%) of the 28 studies featured interventions performed within less than 2 weeks but showed an effect on the mean 24-h glucose level [61]. However, if a telephonebased coaching program for patients with diabetes [62] and CGM users [63] over a period of 6 months prompted a significant improvement in anthropometric indices, dietary habits and hypoglycemia, our 4-week intervention period may have been extremely short. Second, we used only a few biochemical indices as health outcome variables; other indicators, such as blood glucose level, high-density lipoprotein cholesterol level and low-density lipoprotein cholesterol level, were not included. Third, having a small sample size recruited from a single hospital can potentially limit the external validity. Despite attempts to recruit a robust sample within a period of 5 months for this study, several eligible patients withdrew prior to the collection of data. Therefore, concerted and coordinated efforts by various scientists and clinicians involved in this study may be necessary to improve participation [64]. Fourth, confounding variables were not assessed; for example, patient’s age at the time of disease onset masked or minimized the significance of some of our results. A systematic review found that age of T2DM onset was related to a higher risk of mortality and vascular complications [65]; however, additional factors related to age at onset may interfere with the interpretation of these results. Fifth, we did not examine the longer-term effects of our intervention because of situational limitations. Hence, future studies with more participants should be conducted to validate the intervention programs within a longer period.

## 5. Conclusions

This study examined the effects of a dietary coaching program on behavioral skills and health outcomes in patients with diabetes or prediabetes who implemented CGM. Within the experimental group, the dietary coaching program led to a change in male participants’ thigh circumference and in female participants’ eating self-efficacy. Based on the results of our study, these positive changes can relieve insomnia, much in the way that decreased waist circumference can reduce insulin resistance. Taken together, these findings suggest that non-contact dietary coaching programs can enhance behavioral skills, such as eating self-efficacy, as well as health outcomes, such as thigh circumference, even when conducted within a short-term period. In a clinical setting, health coaching by a healthcare professional, even if conducted through a non-contact method, should be provided to promote the effectiveness of CGM in diabetic and prediabetic patients. These outcomes not only indicate the positive effect of the dietary coaching program, but may also bring about additional health improvements either directly or indirectly. However, further research is needed to develop advanced dietary coaching programs and assess their efficacy within a long-term period.

## Figures and Tables

**Figure 1 healthcare-11-00252-f001:**
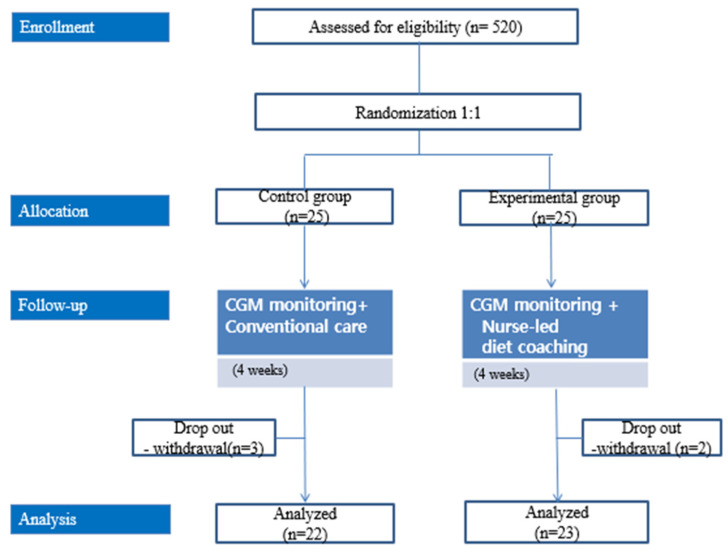
Study flow.

**Figure 2 healthcare-11-00252-f002:**
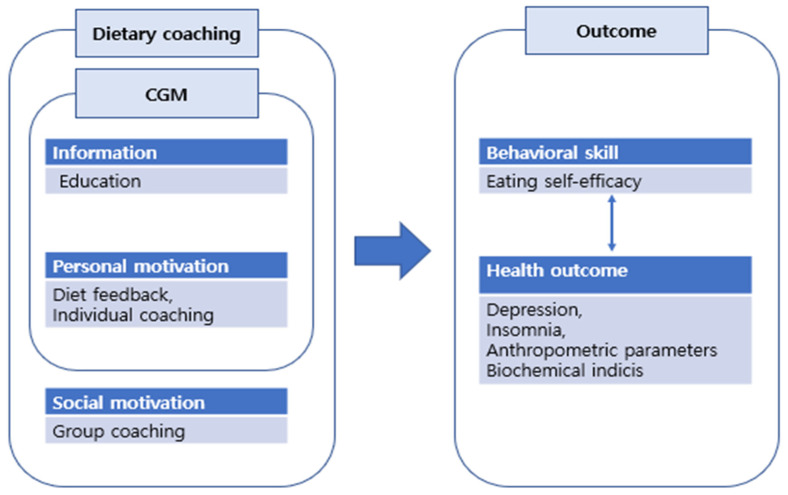
Conceptual framework including the research outcome measures.

**Table 1 healthcare-11-00252-t001:** Homogeneity tests of the participants (*N* = 45).

Characteristic	Categories	Con. (*n* = 22)	Exp. (*n* = 23)	t (*p*)
		M ± SD or *n* (%)		
Age (Year)		49.80 ± 10.42	47.16 ± 10.60	0.89 (0.379)
Sex	Male	11 (50.0)	6 (26.1)	4.16 (0.79)
	Female	11 (50.0)	17 (73.9)	
Height (cm)		167.52 ± 8.36	164.90 ± 8.43	1.10 (0.275)
Weight (Kg)		77.07 ± 14.96	78.09 ± 15.47	0.24 (0.407)
Behavioral skills	Eating self-efficacy	6.88 ± 1.76	6.17 ± 2.18	0.11 (0.240)
Health outcome	Depression	0.66 ± 0.71	0.43 ± 0.55	1.18 (0.243)
	Insomnia	2.07 ± 1.06	1.95 ± 0.75	0.45 (0.657)
	BMI	26.82 ± 3.47	29.01 ± 5.09	−1.68 (0.100)
	Waist circumference	95.80 ± 9.80	96.20 ± 9.26	−0.14 (0.886)
	Thigh circumference	50.52 ± 6.22	51.87 ± 6.00	−0.74 (0.464)
	HbA1c	6.97 ± 1.04	6.78 ± 1.09	0.58 (0.562)
	ApoB/ApoA	2.44 ± 8.39	0.56 ± 0.21	0.41 (0.687)
	HOMAIR	6.68 ± 8.04	4.65 ± 4.73	1.07 (0.289)

BMI, body mass index; HbA1c, glycated hemoglobin; ApoB, apolipoprotein; ApoA, apolipoprotein; HOMAIR, homeostatic model assessment for insulin resistance.

**Table 2 healthcare-11-00252-t002:** Effects of non-contact dietary coaching program in the male participants (*n* = 17).

Characteristics	Categories	Con. (*n* = 11)	Exp. (*n* = 6)	Z (*p*)	Effect Size
		Differences between Pre-Post Test			
		M ± SD			
Behavioral skills	Eating self-efficacy	0.20 ± 1.29	0.29 ± 1.01	−0.30 (0.763)	0.006
Health outcome	Depression	−0.14 ± 0.50	−0.08 ± 0.20	−0.20 (0.839)	0.003
	Insomnia	0.41 ± 0.62	−0.04 ± 0.37	−1.34 (0.180)	0.112
	BMI	−0.60 ± 0.76	−1.72 ± 1.46	−1.56 (0.118)	0.152
	Waist circumference	−2.09 ± 5.52	−4.50 ± 4.68	−1.01 (0.314)	0.064
	Thigh circumference	−1.37 ± 3.73	2.37 ± 2.68	−2.02 (0.044)	0.255
	HbA1c	−0.13 ± 0.31	−0.07 ± 0.29	−0.31 (0.754)	0.006
	ApoB/ApoA	−0.06 ± 0.18	0.00 ± 0.10	−0.30 (0.763)	0.006
	HOMAIR	11.08 ± 108.05	8.66 ± 13.45	−0.10 (0.920)	0.001

BMI, body mass index; HbA1c, glycated hemoglobin; ApoB, apolipoprotein; ApoA, apolipoprotein; HOMAIR, homeostatic model assessment for insulin resistance.

**Table 3 healthcare-11-00252-t003:** Effects of non-contact dietary coaching program in the female participants (*n* = 28).

Characteristics	Categories	Con. (*n* = 11)	Exp. (*n* = 17)	Z (*p*)	Effect Size
		Differences between Pre-Post Test			
		M ± SD or *n* (%)			
Behavioral skills	Eating self-efficacy	−0.75 ± 1.46	0.79 ± 1.29	−2.66 (0.008)	0.262
Health outcome	Depression	0.05 ± 0.47	0.09 ± 0.44	−0.44 (0.660)	0.007
	Insomnia	0.20 ± 0.66	0.07 ± 0.53	−0.41 (0.685)	0.006
	BMI	−0.22 ± 0.68	−0.30 ± 1.10	−0.14 (0.887)	0.001
	Waist circumference	−0.06 ± 4.45	−3.08 ± 5.12	−1.28 (0.200)	0.013
	Thigh circumference	0.45 ± 1.27	−0.05 ± 2.41	−1.63 (0.104)	0.098
	HbA1c	−0.18 ± 0.25	−0.09 ± 0.43	−1.02 (0.308)	0.039
	ApoB/ApoA	−3.64 ± 11.87	0.03 ± 0.17	−1.53 (0.126)	0.087
	HOMAIR	−5.41 ± 37.90	8.18 ± 49.41	−0.23 (0.815)	0.002

BMI, body mass index; HbA1c, Glycated hemoglobin; ApoB, apolipoprotein; ApoA, apolipoprotein; HOMAIR, homeostatic model assessment for insulin resistance.

**Table 4 healthcare-11-00252-t004:** Relationships between sex, group and differences of behavioral skill and health outcome (*N* = 45).

	1	2	3	4	5	6	7	8	9	10
	r (p)									
1. Group	1.00									
2. Sex	0.25 (0.103)	1.00								
3. Change of eating self-efficacy	0.34 (0.023)	−0.02 (0.913)	1.00							
4. Change of depression	0.10 (0.504	0.21 (0.164)	−0.25 (0.098)	1.00						
5. Change of insomnia	−0.23 (0.123)	−0.11 (0.483)	−0.35 (0.018)	0.23 (0.135)	1.00					
6. Change of BMI	−0.12 (0.423)	0.33 (0.026)	−0.04 (0.801)	−0.18 (0.249)	0.02 (0.908)	1.00				
7. Change of waist circumference	−0.24 (0.117)	0.10 (0.508)	−0.26 (0.086)	0.05 (0.764)	0.14 (0.361)	0.25 (0.092)	1.00			
8. Change of thigh circumference	0.05 (0.750)	0.17 (0.267)	0.09 (0.539)	−0.10 (0.509)	−0.35 (0.017)	0.10 (0.514)	0.30 (0.048)	1.00		
9. Change of HbA1c	0.11 (0.483)	−0.03 (0.857)	0.11 (0.492)	−0.28 (0.068)	0.15 (0.312)	0.12 (0.436)	0.27 (0.077)	0.22 (0.147)	1.00	
10. Change of ApoB/ApoA	0.16 (0.290)	−0.12 (0.453)	0.06 (0.686)	−0.00 (0.996)	0.11 (0.472)	−0.21 (0.160)	−10 (0.527)	−0.06 (0.720)	0.22 (0.148)	1.00
11. Change of HOMAIR	0.04 (0.774)	−0.06 (0.706)	0.02 (0.896)	−0.16 (0.297)	−0.12 (0.440)	0.19 (0.203)	0.35 (0.020)	0.12 (0.440)	0.17 (0.255)	0.05 (0.760)

## Data Availability

The datasets generated and analysed during the current study are not publicly available due but are available from the corresponding author on reasonable request.

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
