# Peer review of "Effectiveness of Non-Contact Dietary Coaching in Adults with Diabetes or Prediabetes Using a Continuous Glucose Monitoring Device: A Randomized Controlled Trial"

_healthcare, 2023, doi:10.3390/healthcare11020252_

Round 1
Reviewer 1 Report (Previous Reviewer 1)
The manuscript is improved, and the authors made corrections according to suggestions…But, I need more clarifications regarding CGM.
Please add more info about is CGM (how long did they used, how many sensors, how many scans per day and why did not authors analyse data from CGM), please clarify
Author Response
|
We have added more information about the CGM in the intervention section. |
“The FreeStyle Libre CGM system consists of a sensor that is attached to the back of the upper arm and inserted below the skin [15]. Each sensor is calibrated and can be used for up to 2 weeks. A user device or smartphone with near-sensor communication capability can be shown at any time to scan the sensor. The device or smartphone monitor shows the current glucose level, the glucose level trend, and a graph of the glucose levels obtained in the last 8 h [16]. The CGM results including the current glucose level, the glucose trend, and the graph were not used as one of the biochemical indices in this study. This is because it was not possible to accurately analyze the effects of this program as the participants’ diet and medication intake time were unknown.”

Reviewer 2 Report (Previous Reviewer 2)
The authors addressed the required points adequately. There is need for moderate English editing for gramatical and scientific soundness.
Author Response
We got an English editing for grammatical and scientific soundness. Thank you.

Reviewer 3 Report (Previous Reviewer 4)
I find the authors revised all reviewers' comments.
Author Response
We got an English editing for grammatical and scientific soundness. Thank you.

This manuscript is a resubmission of an earlier submission. The following is a list of the peer review reports and author responses from that submission.
Round 1
Reviewer 1 Report
The manuscript “Effectiveness of Non-contact Dietary Coaching in adult with Diabetes or Prediabetes using a Continuous Glucose Monitoring Device: A Randomized Controlled Trial” is 4 week RCT aimed to evaluate the effectiveness of dietary coaching, along with continuous glucose monitoring (CGM), in patients with diabetes or prediabetes to improve behavioural skills and health outcomes. Forty-five patients with diabetes or prediabetes who used a CGM device in two outpatient department of a tertiary hospital were included. Participants in the control group were provided with conventional care for over 4 weeks. Dietary coaching was provided to members of the experimental group during the same period. After intervention, eating self-efficacy showed significantly greater improvements in the experimental as compared with the control group. The are minor issues to be clarified:
Comments to the Authors:
1. Please, put the study design in the Abstract section
2. In methodology section: please add, are the hospitals public or private, and is health insurance covered by participants or not.
3. Methodology section…Did all subjects with prediabetes had CGM, please add explanation
Reviewer 2 Report
The manuscript entitled “ Effectiveness of Non-contact Dietary Coaching in adult with Diabetes or Prediabetes using a Continuous Glucose Monitor-3 ing Device: A Randomized Controlled Trial” The manuscript is valuable and well written but some minor concerns are listed below , I hope they will improve the manuscript.
- The abstract needs substantial revision and please write the results and the details of the biochemical indices found.
- In the introduction: you should mention about the studies have performed lifestyle modification interventions in T2DM.
- The reader should read full idea about the term dietary coaching and its role in different states to modify the health state at the beginning of the introduction not after that.
- How could you estimate the prediabetic states clinically and how could you adjust the cases of different patients at the same base line to start the experiment , are there sharp criteria for that.
- How could T2DM modify the behavioral skills in patients without any dietary coaching and how could this be evaluated?
- The limitations of the study should be mentioned.
- Conclusion needs revision to cope the aim of the study.
Reviewer 3 Report
First of all, I would like to acknowledge that this is an interesting research and that I have enjoyed reading and analysing it. I also would like to say that it has potential and interest, although it requires some important improvements in its current format, so that the work can be publishable and of interest to the scientific community. I suggest the following improvements:
The abstract would be much improved if it were structured and its headings were identified: objectives, material and methods, results and conclusion. Also if some essential numerical results were provided, beyond the result of the hypothesis test and the level of significance.
In the introduction I think it would also be necessary to go into more depth on the most effective interventions in lifestyle modification, expanding the literature cited, with more background on experimental studies that analyse the interventions that have the greatest effect.
There are only three references on the impact of CGM on adherence in patients with DM, and it seems that they barely explore patients with Type 1 DM. With so little evidence of its efficacy in patients with Type 2 DM, it is perhaps risky to argue how beneficial its application would be in this type of patient. The starting position could be improved with more literature support, perhaps by analysing the effect of continuous monitoring devices even for other parameters in other pathologies. I encourage a more thorough search on this issue.
I also believe that the authors should make an effort to relate the CGM procedure to lifestyle modification interventions, including the IMB model or others. These two basic elements of the study are addressed in the introduction in a very isolated way.
I also believe that the introduction does not need to discuss the "pilot study" nature of the work, but that this aspect should be addressed exclusively in the introduction.
In the section on material and method, I think there are important gaps to be clarified. The first is precisely the reason for the "pilot" nature of the study. Besides that, I think there are other exclusion criteria that perhaps should have been considered, such as cognitive impairment. The flow chart should follow more strictly the recommendations of the CONSORT statement. At no point was it considered to stratify the sample according to the type of diabetes? As this seems to be a differential factor in previous literature, it could have been taken into account when forming the sample. In any case, the type of sampling is not clearly specified (is it random? Convenience? Accidental?). This is basic and the possible selection bias of the study should be reconsidered.
Regarding the allocation of participants to each of both groups (control and intervention), although it is mentioned that it is random, more detail on the process is needed.
Be careful because it uses an inappropriate bibliographic citation style. I think figures 2 and 3 could be sent to an appendix, as although they are illustrative they interrupt the reading too much.
What the intervention consists of is described in great detail, but it is clear from the description that the control group receives absolutely no comparative intervention, and I have doubts as to whether this is not questionable from an ethical point of view. Furthermore, in addition to the hospital's authorisation, given the experimental nature of the study, approval by an ethics committee, with its corresponding number, is required.
The analysis performed is exclusively bivariate and I believe that it would enrich the study much more if a mutlivariate analysis were performed, given the variability of variables for the measurement of outcome.
The results should be better described in the narrative, as its description is very poor and limited. The tables are confusing and I think that basic information such as the level of significance obtained (exact p-value) or the effect size, which are not provided, should be added. I think this section needs more effort in terms of presentation.
I also believe that an effort should be made to better discuss the limitations of the study, which, although some of them are identified, have been overlooked, such as the nature of the sample, the impossibility of performing a "blinding" process in the allocation of participants, the small sample size, the possible biases in the measurement of the outcome, or the confounding effect that may be produced by variables not addressed such as the type of DM or the age of the patient. I believe that the external validity of the study needs to be relativised and better justified.
By the way, in addition to the proposed conclusions, it would be very wise to add some reflection on the scope of the study and its prospective for clinical practice.
The analysis performed is exclusively bivariate and I believe that it would enrich the study much more if a mutlivariate analysis were performed, given the variability of variables for the measurement of outcome.
The results should be better described in the narrative, as its description is very poor and limited. The tables are confusing and I think that basic information such as the level of significance obtained (exact p-value) or the effect size, which are not provided, should be added. I think this section needs more effort in terms of presentation.
I also believe that an effort should be made to better discuss the limitations of the study, which, although some of them are identified, have been overlooked, such as the nature of the sample, the impossibility of performing a "blinding" process in the allocation of participants, the small sample size, the possible biases in the measurement of the outcome, or the confounding effect that may be produced by variables not addressed such as the type of DM or the age of the patient. I believe that the external validity of the study needs to be relativised and better justified.
By the way, in addition to the proposed conclusions, it would be very wise to add some reflection on the scope of the study and its prospective for clinical practice.
The analysis performed is exclusively bivariate and I believe that it would enrich the study much more if a mutlivariate analysis were performed, given the variability of variables for the measurement of outcome.
The results should be better described in the narrative, as its description is very poor and limited. The tables are confusing and I think that basic information such as the level of significance obtained (exact p-value) or the effect size, which are not provided, should be added. I think this section needs more effort in terms of presentation.
I also believe that an effort should be made to better discuss the limitations of the study, which, although some of them are identified, have been overlooked, such as the nature of the sample, the impossibility of performing a "blinding" process in the allocation of participants, the small sample size, the possible biases in the measurement of the outcome, or the confounding effect that may be produced by variables not addressed such as the type of DM or the age of the patient. I believe that the external validity of the study needs to be relativised and better justified.
By the way, in addition to the proposed conclusions, it would be very wise to add some reflection on the scope of the study and its prospective for clinical practice.
Reviewer 4 Report
Ahn et al want to evaluate the effectiveness of dietary coaching, along with continuous glucose monitoring (CGM), in patients with diabetes or prediabetes to improve behavioral skills and health outcomes. Biochemical variables and so on were determined. It's some interesting. But I think:
1. The content is very simple, it doesn't enough to publish in this status.
2. All indexes must be analysised from male and female separately, for sex difference.
3. Some correlation analysis should be done. Thus, some more information can be given.